# Cadmium, Lead, Copper, Zinc, and Iron Concentration Patterns in Three Marine Fish Species from Two Different Mining Sites inside the Gulf of California, Mexico

**DOI:** 10.3390/ijerph18020844

**Published:** 2021-01-19

**Authors:** Elisa Serviere-Zaragoza, Salvador E. Lluch-Cota, Alejandra Mazariegos-Villarreal, Eduardo F. Balart, Hugo Valencia-Valdez, Lia Celina Méndez-Rodríguez

**Affiliations:** Centro de Investigaciones Biológicas del Noroeste (CIBNOR), Calle IPN 195, La Paz, Baja California Sur 23096, Mexico; serviere04@cibnor.mx (E.S.-Z.); slluch@cibnor.mx (S.E.L.-C.); amaza04@cibnor.mx (A.M.-V.); ebalart04@cibnor.mx (E.F.B.); hughvalen@gmail.com (H.V.-V.)

**Keywords:** *Balistes*, *Kyphosus*, *Stegastes*, trace elements, stable isotopes, phosphorite mining, copper mining

## Abstract

In the Gulf of California; mineral deposits have contributed to high metal contents in coastal environments. This study examined cadmium; lead; copper; zinc; and iron contents in three fish species; *Kyphosus vaigiensis* (herbivore), *Stegastes rectifraenum* (omnivore), and *Balistes polylepis* (carnivore) at two mining sites. Metal concentrations were analyzed by atomic absorption spectrophotometry and stable nitrogen and carbon isotopes were estimated using mass spectrophotometry. Also, we assessed the risk to human health from the consumption of these three species based on permissible limits; although only two of them (*Kyphosus* and *Balistes*) are consumed as food. Metal concentrations differed among fish species; except for iron. The highest concentrations of metals were not always recorded in the species at the highest trophic level; i.e., *Balistes*. The highest concentrations (dry weight) recorded were cadmium (0.21 ± 0.03 µg g^−1^) and lead (1.67 ± 0.26 µg g^−1^), in *S. rectifraenum*; copper (1.60 ± 0.49 µg g^−1^) and zinc (67.30 ± 8.79 µg g^−1^), in *B. polylepis*; and iron (27.06 ± 2.58 µg g^−1^), in *K. vaigiensis*. Our findings show that each element accumulates differently in particular marine organisms; depending on the physiology of the species and the biogeochemistry of its habitat; which in turn is affected by the anthropogenic activities in adjacent areas. No risk of heavy metals toxicity is expected from the human consumption of the species and sites studied

## 1. Introduction

Trace elements can cause harmful health effects when they accumulate in organisms at concentrations above those required for the metabolic functions [1]. Zinc, copper, and iron are essential elements involved in different metabolic processes, and most organisms have biochemical mechanisms to regulate the amount of these elements in their cells [1]. Lead and cadmium are non-essential elements that compete with essential elements for enzyme sites. Although less than 20% of these elements are assimilated in the human body, their half-life in human tissues is around 30 years [2,3]. When in excess, metals have been associated with serious effects to both human and animal health. For example, high intake levels of copper, zinc, and lead have been related to Alzheimer’s disease; zinc and iron, with Parkinson’s disease; cadmium may induce kidney dysfunctions, osteomalacia, and reproductive deficiencies, among others [4,5].

Local mineral deposits in the environment may contribute to increase the content of one or more chemicals in sediments relative to the levels considered as typical of the Earth’s crust [6]. The chemical composition and bioavailability of trace metals in the food consumed by fish and the water column where they thrive are influenced by the local geochemistry and anthropogenic activities such as mining and agriculture; both may lead to enrichment with high amounts of metallic elements [1]. However, the relationship between trace metals concentration in fish tissue and mining activities has not been fully addressed in the scientific literature, especially regarding phosphorite mining and the comparisons between sites exploiting different minerals. The available reports suggest variable bioaccumulation patterns among fish sharing the same location [7,8]. In a site polluted by phosphorite mining on the coast of Togo, the highest metal concentration observed in fish were cadmium (1.68 µg g^−1^) in *Chloroscombrus chrysurus*, lead (8.49µg g^−1^) in *Galeoides decadactylus*, and zinc and iron (1.48 µg g^−1^ and 2.99 µg g^−1^, respectively) in *Ilisha africana* [8]. In another study, the grass carp *Ctenopharyngodon idellus* thought to be affected by copper mining in the district of Purple Mountain, China, for over 30 years, recorded concentrations (dry weight) of up to 8.5 ± 0.75 µg g^−1^ of copper, 27.7 ± 2.7 µg g^−1^ of zinc, and 0.705 ± 0.155 µg g^−1^ of lead, and no detectable concentrations of cadmium [7]. In fish from a site affected by the collapse of an iron mining dam in the Rio Doce, Brazil, the highest concentration (dry weight) of lead and copper (8.55 and 1.15 µg g^−1^, respectively) were recorded in *Eugerres brasilianus*, and the highest concentration of zinc (477 µg g^−1^) was recorded in *Genidens genidens*. Cadmium levels were below the limit of detection in fish [9]. Further research is needed to understand whether there is a detectable and consistent effect of mining activities on the bioaccumulation of trace metals in fish inhabiting nearby coastal waters.

Deposits of various minerals are located in the Gulf of California, some of which have been exploited through mining operations at sites adjacent to the coast [10]. Two mining sites of particular interest because of nearby human settlements and coastal fishery activities are Santa Rosalía (STR), at the central part of the gulf, and Bahía de La Paz (LAP), near the tip of the Baja California Peninsula, both located on the west coast of the Gulf of California, México (Figure 1). In STR, copper extraction operations were carried out since the second half of the 19th century to the late 20th century [11]. Tons of wastes from these mining operations were deposited over the years on the beach and directly into the marine environment. The copper content on the beaches of STR (up to 30,380 µg g^−1^) is higher than levels recorded in industrialized areas of Russia [12]. Iron and zinc deposits are also found in this zone [10], and high lead concentrations (up to 2100 µg g^−1^) have been found in sediments [12]. The second case addressed here regards the current phosphorite extraction operations in San Juan de la Costa, inside LAP, mainly to obtain phosphorus for agricultural fertilization [10]. Besides phosphorus, phosphorite may also contain zinc, lead, and cadmium [13]. Reports of cadmium in LAP sediments indicate concentrations above those considered typical of the Earth’s crust (>0.1 µg g^−1^) [6].

The coastal areas off STR and LAP are home to macroalgae beds with the capacity to remove trace elements suspended in the water column [11], which may potentially be transferred to herbivores foraging on them. Depending on the chemical presentation of these elements, they may or may not be bioaccumulated by organisms feeding on macroalgae and transferred to higher trophic levels [14]. The dominant components of the marine flora dwelling on the coastal marine environments across the Gulf of California, including STR and LAP, are brown algae of the genus *Sargassum*. These algae cover large areas, providing habitat, shelter, and food to numerous species of invertebrates and fish [15], including the blue-bronze chub *Kyphosus vaigiensis* (Quoy and Gaimard, 1825), the Cortez damselfish *Stegastes rectifraenum* (Gill, 1862), and the fine scale triggerfish *Balistes polylepis* (Steindachner, 1876). *S. rectifraenum* is highly appreciated as an ornamental fish for aquarists, while the other two species are captured by artisanal fishers for direct human consumption [15].

Almost all members of the Kyphosidae family are herbivorous [16]. Studies on the diet of *K. vaigiensis* confirm that this species is a functional herbivore that feeds primarily on macroalgae, including phaeophytes, chlorophytes, and rhodophytes, and almost no animal material [17,18] as other herbivorous species of *Kyphosus*, such as *K. bigibbus*, *K. cinerascens*, *K. sydneyanus*, *K. azureus*, and *K. elegans* (the latter two also from the Gulf of California) [17,19,20,21]. *S. rectifraenum* (family Pomacentridae) was first described as an herbivore [22], but later studies have reported an omnivorous diet consisting of turf, zoobenthos, and plankton [23,24]. *B. polylepis* (Balistidae) has been classified as predominantly carnivorous, although some algae are consumed occasionally as incidental food [25,26]. The fact that most records of these three fish species are from rocky inshore reefs dominated by *Sargassum* along the Gulf of California [15], suggests that their accumulation of metals is linked to local conditions.

There is information about cadmium, lead, copper, zinc, and iron from *K. vaigiensis*, *K. cinerascens* and *K.* sp. sampled in China, Australia, or New Caledonia (Table 1). No information on trace metals has been reported for *S. rectifraenum*, but there are reports available for other species of this same genus, such as *S. fasciolatus* [27]. As regards the genus *Balistes*, there is information on *B. capriscus* from Mediterranean Sea [28] and on *B. polylepis* from the eastern coast of the Gulf of California and the Pacific coast, but not from the west coast where the two formerly mentioned sites of mining influence are located (Table 1). The objective of this study was to evaluate the cadmium, lead, copper, zinc, and iron concentrations in the muscle of three fish species with different feeding habits inhabiting coastal waters adjacent to two different mining facilities, one for copper and another for phosphorite, and to test whether the type of mining determines the metal concentration in fish muscle. To our knowledge, this is the first report comparing different mining activities in the area. To evaluate the relative importance of the type of mining activity in determining the metal concentrations in fish muscle, we also evaluated other factors, including the time of the year, the trophic niche, and the fish size. The potential risk to human health from the consumption of fish species was assessed based on international guidelines.

## 2. Materials and Methods

### 2.1. Study Area

Sampling was conducted on two locations affected by mining activities, Santa Rosalía (STR) and Bahía de La Paz (LAP), Baja California Sur, on the west coast of the Gulf of California (Figure 1). This coastline is mostly rocky, with scattered sandy stretches and a narrow shelf, and with virtually no drainage from rivers due to sub-desert climate conditions. Atmospheric forcing over the Gulf of California is strongly seasonal; weak southeasterly winds blow through the summer and stronger northwesterly ones during winter, mostly along the gulf axis [38]. Rainfall takes place mostly during the summer [39], when tropical storms and hurricanes produce heavy rainfall and intense water and sediment runoff into the basin [40]. Along the peninsular coast, and particularly near STR, the southerly summer winds cause coastal upwelling events that, together with tidal forcing, result in water column mixing. LAP is the largest coastal body in the southwestern Gulf of California and is characterized by its high diversity and abundance of fish species [41]. Biological enrichment is particularly high due to intense local upwelling events generated by local seasonal wind patterns, topographic configuration, and water exchange between the bay and the gulf [42]. In STR, “El Boleo” mine operated intermittently since the second half of the 19th century up until 1984 [12] and includes copper and zinc deposits. LAP has one of the largest phosphorite deposits worldwide [43], which has been exploited for nearly 30 years [10].

### 2.2. Sampling

The three fish species studied, namely *Kyphosus vaigiensis*, *Stegastes rectifraenum*, and *Balistes polylepis*, were caught in STR and LAP by free diving using a Hawaiian-type harpoon. In STR, three sites were sampled during the dry (May–June 2015) and rainy (September–October 2015) seasons: Punta Gorda (PGO; 27°32′6.6″ N and 112°21′15.4″ W), Las Cuevitas (CUE; 27°24′35.05″ N and 112°58′7.499″ W), and Los Frailes (FRA; 27°18′32″ N and 112°13′58.101″ W) (Figure 1). Las Cuevitas is located at one end of the STR dock, adjacent to the mining operation area. Punta Gorda and Los Frailes are located 10 km north and 3 km south of Las Cuevitas, respectively. In LAP, two sites were sampled during the dry and rainy seasons: Las Ánimas (ANI; 24°50.51′30.99″ N and 110°41.3′22.99″ W) and San Juan de la Costa (SJC; 24°22.26′15.99″ N and 110°41.3′22.99″ W). A phosphorite mine is located at San Juan de la Costa; Las Animas is a fishing village located 14.1 km north of San Juan de la Costa that comprises just a few houses (Figure 1).

A total of 70 specimens of *Kyphosus vaigiensis* (43 from STR; 27 from LAP), 68 of *Stegastes rectifraenum* (40 from STR; 28 from LAP), and 61 of *Balistes polylepis* (33 from STR; 28 from LAP) were captured and transported on ice to the laboratory. The total length (±1 mm) of each fish was recorded. Afterward, fish were dissected, and a sample of white muscle from the anterior dorsal region was excised from each specimen. Muscle samples were rinsed with distilled water and dried in Petri dishes in a dry-heat oven for 24–48 h at 75 °C to constant weight. Then, tissues were ground using a Wig-L-Bug (model 30) electro-mechanical stirrer until a homogeneous powder was obtained. Each sample was transferred to a separate vial and stored in a drying chamber until tested for trace elements. A subsample of muscle from each specimen was saved for isotopic analysis.

### 2.3. Analyses of Trace Metals

Samples of 0.5 g of muscle tissue were digested with nitric acid and hydrogen peroxide in a microwave oven (Mars 5x, CEM, Matthew, NC, USA). After acid digestion, 1 mL of hydrochloric acid (HCl) was added to each sample and the volume was brought to 50 mL with deionized water. Cadmium (Cd), lead (Pb), copper (Cu), zinc (Zn), and iron (Fe) concentrations were determined by atomic absorption spectrophotometry (GBS Scientific AVANTA, Dandenong, Australia) using an air/acetylene flame [44]. Each sample was tested in triplicate. High-purity reagents were used in all cases; blanks were analyzed in parallel to validate the efficiency of the method [45]. A standardized reference material (DORM-2, National Research Council of Canada; Ottawa, ON, Canada) was also analyzed in each sample run as a quality control. The analysis of metal content yielded recovery values ranging from 93% to 116% for the entire process. The detection limits and the quantification limits (µg g^−1^) respectively were as follows: Cd: 0.01 and 0.02; Pb: 0.07 and 0.10; Cu: 0.017 and 0.020; Zn: 0.021 and 0.060; Fe: 0.65 and 1.35.

### 2.4. Analyses of Stable Isotopes

Stable isotope ratios of nitrogen and carbon are powerful tools for evaluating the trophic structure and energy flow across aquatic ecosystems. The δ^15^N of an organism typically shows a 3.4 ‰ (±1‰) increase at each higher trophic level; thus, it can be used to determine the trophic position of an organism [46]. In contrast, δ^13^C changes little as carbon moves through the food web and can be used to evaluate the ultimate sources of energy for an organism [47]. Samples from the 199 specimens were combusted to N_2_ and CO_2_ in tin capsules using a PDZ Europe ANCA-GSL elemental analyzer interfaced to a PDZ Europe 20–20 isotope ratio mass spectrometer (Sercon Ltd., Cheshire, UK) at the Stable Isotope Laboratory, University of California Davis. It was not necessary to account for lipids in fish samples, as the average C:N ratio of all samples (C:N = 3.3 ± 0.1) was below 3.5 [48]. Nitrogen isotope ratios (δ^15^N) are reported in parts per thousand (‰) relative to N_2_ in air, while carbon isotope ratios (δ^13^C) are reported relative to Pee Dee Belemnite (PDB) using the following equation:(1)δX‰ = ⟨RsampleRstandard−1⟩ × 1000
where X = ^15^N or ^13^C and R = ^15^N/^14^N or ^13^C/^12^C. Samples were analyzed interspersed with several replicates of at least four different laboratory reference materials. These reference materials were previously calibrated using international reference materials, including IAEA-600, USGS-40, USGS-41, USGS-42, USGS-43, USGS-61, USGS-64, and USGS-65. The standard deviation was ±0.3 ‰ for ^15^N and ±0.2 ‰ for ^13^C.

### 2.5. Statistical Analysis

Before the statistical tests, to each sample that presented an undetectable concentration of some metal it was assigned a value that corresponds to half the detection limit of such metal [49]. Metal concentrations in fish muscle were log-transformed before performing the statistical analyses to meet the assumptions of normality based on the Shapiro-Wilk test [50,51]. One-way ANOVA analyses were used to assess significant differences in metal concentrations between location (Santa Rosalía, La Paz), also between seasons (dry, rainy) within each location and in specimens length, metal concentration and stable isotopes between species (*K. vaigiensis*, *S. rectifraenum*, *B. polylepis*). ANOVA were followed by the *post hoc* Tukey tests [51].

### 2.6. Multiple Categorial Regression

Multiple categorial regression models of the general form:Metal = *a* + *b*(location) + *c*(season) + *d*(Sp1) + *e*(Sp2) + *f*(L) + *g*(δ^13^C) + *h*(δ^15^N)(2)
were computed for each metal, where *a* is the intercept, *b* takes positive values for STA and negative for LAP, *c* takes positive values for the rainy season and negative for dry, *d* is the coefficient of the dummy variable Sp1 and takes positive values for *K. vaigiensis*, and negative for others, *e* is positive for *S. rectifraenum* and negative for *B. polylepis*, *f* is the partial coefficient of normalized standard fish length, and *g* and *h* are the coefficients for the stable isotopes. Before the analyses, fish length was normalized by species, and the data were arranged according to the categorical variable’s location, season, and species, and also according to the continuous variables fish standard length and isotopic readings (δ^13^C, δ^15^N). The multiple categorial regression was applied to evaluate and compare the relative weight of each of the factors in determining the concentration of the different elements in fish muscle. The models were fitted in the R statistical package using a significance level of *p* < 0.05.

### 2.7. Health Risk

One of our objectives was to compare the results obtained in this study with standards and results of previous works and assess the potential health risk of human consumption of muscle of the three species studied. To this end, element concentrations were converted from dry weight (dw) to fresh weight (fw) as follows:Element fw = Element dw × ((100 − % moisture) 100)(3)
using the percentage of moisture in muscle tissue of each fish species (range 73.8–79.4%) [52]. The Estimated Daily Intake (EDI, mg trace element kg^−1^ BW day^−1^) was calculated as follows [53]:EDI = (Cm × CR)/BW(4)
where Cm = mean concentration (trace metals) in muscle tissue, expressed as a fresh weight (µg g^−1^); CR = Mean per-capita daily consumption rate of fish muscle (30 mg day^−1^); BW = mean body weight of an adult person (65 kg) [54]. From a nutritional standpoint, we used the recommended daily intake (RDI [55]) for iron (8 mg day^−1^), zinc (11 mg day^−1^), and copper (0.9 mg day^−1^) to assess the contribution of the estimated EDI for these elements. No RDI has been set for cadmium and lead since these metals are considered toxic to humans. To assess the potential toxicological risk of the intake of the elements studied through the consumption of muscle from each of the three species, each EDI was compared with its respective reference oral dose (RfD), namely: cadmium: 1 μg kg^−1^ bodyweight day^−1^; copper: 40.0 μg kg^−1^ bodyweight day^−1^; iron: 700 μg kg^−1^ bodyweight day^−1^, and zinc: 300 μg kg^−1^ bodyweight day^−1^ [56]. An RfD value has not been set for lead; USEPA and IRIS EPA consider lead to be a special case because of the difficulty in identifying the classic “threshold” needed to establish an RfD [3]. Radial plots were built where the EDI for each of the metals, species, and locations was expressed as a proportion (%) of the RfD or the total daily intake. In the case of lead, for which no RfD value has been established, we considered a value of 4 μg kg^−1^ bodyweight day^−1^ [34,54,57].

## 3. Results

Of the 70 specimens of *K. vaigiensis*, 38 (17 from STR, 21 from LAP) showed undetectable cadmium levels (<0.01 µg g−1), 48 (27 from STR, 21 from LAP) had undetectable lead levels (<0.07 µg g−1), and 16 (1 from STR, 15 from LAP) showed undetectable copper levels (<0.017 µg g−1). Of the 68 samples of *S. rectifraenum*, 20 (3 from STR, 17 from LAP) showed undetectable cadmium levels, 34 (23 from STR, 11 from LAP) had undetectable lead levels, and 26 (19 from STR, 7 from LAP) showed undetectable copper levels. Of the 61 *B. polylepis* organisms, 24 (7 from STR, 17 from LAP) showed undetectable cadmium levels, 48 (27 from STR, 21 from LAP) had undetectable lead levels, and 18 (5 from STR, 13 from LAP) showed undetectable levels of copper.

### 3.1. Locations and Seasons

Fish sampled in STR were, on average, smaller than in LAP for *S. rectifraenum* (STR = 11.09 ± 0.16 cm and LAP = 22.97 ± 0.89 cm) and *B. polylepis* (STR = 12.47 ± 0.13 cm and LAP = 26.62 ± 1.05 cm); the opposite occurred for *K. vaigiensis* (STR = 31.82 ± 1.32 cm and LAP = 23.70 ± 1.42 cm). Concerning the metal concentrations between locations, the annual average concentrations of cadmium (0.19 ± 0.03 µg g^−1^) (F_(1,68)_ = 13.64, *p* = 0.000), copper (1.57 ± 0.20 µg g^−1^) (F_(1,68)_ = 6.74, *p* = 0.012), and iron (32.60 ± 3.65 µg g^−1^) (F_(1,68)_ = 8.90, *p* = 0.004) in *K. vaigiensis* fish caught in STR were higher (6.3-, 1.9-, and 1.8-fold, respectively) than concentrations in fish from LAP, while the concentrations of lead (0.59 ± 0.16 µg g^−1^) and zinc (42.09 ± 2.96 µg g^−1^) were 0.6-fold or similar to those recorded in fish from LAP. In the case of *S. rectifraenum*, Cd, Cu, and Fe concentrations were not significantly different in fish between STR and LAP, while the levels of Pb (2.76 ± 0.49 µg g^−1^) (F_(1,66)_ = 14.88, *p* = 0.000) and Zn (23.88 ± 0.77 µg g^−1^) (F_(1,66)_ = 10.87, *p* = 0.002) were higher (3.1- and 1.1-fold, respectively) in fish caught in LAP versus STR. Regarding *B. polylepis*, Zn concentrations (93.72 ± 14.77 µg g^−1^) (F_(1,59)_ = 12.74, *p* = 0.001) were significantly higher (2.6-fold) in samples from STR versus LAP; no significant differences were observed for any of the remaining metals (Table 2).

Table 2 compares metal concentrations between seasons for each species and each location. In the case of *K. vaigiensis*, for fish caught in STR, Cd, Pb, Cu, and Zn levels were not significantly different between dry and rainy season samples, while Fe in fish caught in the dry season (47.70 ± 5.14 µg g^−1^) was higher than in the rainy season (16.78 ± 1.93 µg g^−1^) (F_(1,41)_ = 28.25; *p* = 0.000). In fish from LAP, the concentrations of Cd (0.06 ± 0.03 µg g^−1^)(F_(1,25 )_ = 5.17, *p* = 0.03), Pb (2.08 ± 0.71 µg g^−1^) (F_(1,25)_ = 8.89, *p* = 0.006), Cu (1.71 ± 0.21 µg g^−1^) (F_(1,25)_ = 69.63, *p* = 0.000), and Zn (55.01 ± 4.72 µg g^−1^) (F_(1,25)_ = 19.79, *p* = 0.000) were higher in samples obtained in the dry season than in the rainy season; no significant differences were observed for Fe concentrations. Regarding *S. rectifraenum* caught in STR, significant differences were detected only in Cu concentration, being higher in fish from the dry season (1.05 ± 0.12 µg g^−1^) versus the rainy season (F_(1,38)_ = 76.46, *p* = 0.000). In LAP, Pb (4.16 ± 0.43 µg g^−1^) (F_(1,26)_ = 11.42, *p* = 0.002) and Cu (0.88 ± 0.14 µg g^−1^) (F_(1,26)_ = 10.58, *p* = 0.003) levels were higher in fish caught in the dry season versus the rainy season; no significant differences were observed in the other metals. As for *B. polylepis*, significant differences between seasons were detected in Zn concentrations in fish caught in STR, with higher concentrations in the rainy season (123.54 ± 20.58 µg g^−1^) than in the dry season (F_(1,31)_ = 8.89, *p* = 0.005) and in Pb and Fe concentrations, being higher in fish from dry season (F_(1,31)_ = 12.55, *p* = 0.001 and F_(1,31)_ = 12.38, *p* = 0.001 respectively); no significant differences in metal concentrations were observed in fish caught in LAP (Table 2).

### 3.2. Fish Species

Table 3 compares the general characteristics, body length, metal concentrations, and isotopic measurements for the three fish species. Body length was significantly smaller (*p* < 0.05) in *S. rectifraenum* (11.66 ± 0.14 cm), and no differences were observed between *B. polylepis* (24.65 ± 0.71 cm) and *K. vaigiensis* (28.69 ± 1.08 cm). Metal concentrations of Cd (F_(2,193)_ = 4.32, *p* = 0.015), Pb (F_(2,193)_ = 4.52, *p* = 0.012), Cu (F_(2,193)_ = 3.68, *p* = 0.027) and Zn (F_(2,193)_ = 20.81, *p* = 0.000) and both stable isotopes δ^13^C (F_(2,193)_ = 32.20, *p* = 0.000) and δ^15^N (F_(2,193)_ = 20.95, *p* = 0.000) in fish muscle differed between fish species. The highest mean concentrations recorded were cadmium and lead in *S. rectifraenum,* copper and zinc in *B. polylepis,* and iron in *K. vaigiensis*. *B. polylepis* showed no significant differences in mean cadmium, lead, copper, and iron contents relative to the other two species. As regards stable isotopes, the average value of δ^13^C showed significant differences between the three species, with the lowest values found in *B. polylepis* (−16.25 ± 0.21), followed by *S. rectifraenum* (−14.99 ± 0.19) and *K. vaigiensis* (−13.66 ± 0.27); for δ^15^N, the mean value for *K. vaigiensis* was significantly lower (16.77 ± 0.16) versus *S. rectifraenum* (18.12 ± 0.13) and *B. polylepis* (17.73 ± 0.17).

### 3.3. Multiple Categorical Regression

Multiple linear regression models (Table 4) were run to evaluate the influence of the location, season, fish species, fish size (normalized standard length), food source (δ^13^C), and trophic level (δ^15^N). The only factor that significantly explains the cadmium concentrations observed is the location (Cd in STR). For lead, significantly contributing factors were season (higher Pb in the dry season), fish species (higher Pb in *S. rectifraenum*), and fish size (higher Pb in larger fish). For copper, the location (higher Cu in STR) and season (higher Cu in the dry season). For zinc, the season (higher Zn in the rainy season), fish species (higher Zn in *B. polylepis*), fish size (higher Zn in larger individuals), and food source (negative relationship with δ^13^C); finally, for iron, the factors explaining the concentration are location (higher Fe in LAP), fish size (higher Fe in larger organisms), and trophic level (inverse relationship with δ^15^N).

### 3.4. Health Risk Assessment

The estimated daily intake (EDI) of copper, zinc, iron, and cadmium in humans from the consumption of muscle of each of the fish species studied is shown in Figure 2, along with the nutritional contribution from consuming 30 g per day (210 g per week) of fish by a person of 65 kg. In terms of EDI, *B. polylepis* (Figure 2) was the species that contributes the highest amount of copper (approximately 19% of the recommended daily intake—RDI) and zinc (about 85% of the RDI), while *K. vaigiensis* contributes almost 45% of the RDI of iron. In the case of cadmium, *S. rectifraenum* is the species supplying the highest amount of this element, about 40% of the reference dose (RfD). In the three fish species, the estimated EDI of copper, zinc, iron, and cadmium are below their respective RfD (Figure 2). In the case of lead, only individuals of *S. rectifraenum* from LAP exceeded the established value (Figure 2).

## 4. Discussion

Is there a detectable, consistent effect of mining activities in the bioaccumulation of trace metals in fish, or are local oceanographic and geochemical dynamics the key drivers? Does bioaccumulation depend on the type of minerals exploited or the unique characteristics of each fish species? Our observations reveal that there is no clear pattern of differences in trace metal concentrations between STR (copper mining) and LAP (phosphorite mining); instead, the differences appear to be mostly related to the physiology of the sampled fish species and the local biogeochemical dynamics.

The presence of cadmium in some localities of the Gulf of California has been attributed to natural land-based sources, most likely supplied by weathering and transport from phosphorite-containing rocks [65] that may also contain lead [13]. In the same sites where fish were caught in LAP, the urchins *Tripneustes depressus* (30.9 ± 5.5 µg g^−1^) and *Eucidaris thouarsii* (38.3 ± 2.3 µg g^−1^) had cadmium concentrations nearly 3-fold higher than peak levels in macroalgae [66]. Also, high lead concentrations have been reported in other species (Table 1) from the same sites in LAP; these include concentrations up to 38.6 ± 4.2 µg g^−1^ in the sea urchin *Eucidaris thouarsii* attributed to the consumption of crustose coralline (*Lithophyllum*) or articulated coralline (*Amphiroa*) algae or other organisms such as mollusks, barnacles, filamentous, or turf-forming algae attached to macroalgae or rocks [61,66]. Therefore, we expected to find higher cadmium and lead levels in LAP, linked to phosphorite mining. Instead, the multivariate categorical regression showed that location was associated with cadmium levels, indicating higher Cd in STR, but no significant association of location with lead. As regards the differences in metal content between species and locations, the only species with significant differences in cadmium concentration between sites was *K. vaigiensis*, which is the species at the lowest trophic level of the three fish species studied. Undetectable or very low concentrations of copper (<0.03 µg g^−1^) in muscle have been previously reported [67,68,69]. Concentrations of this and other elements in fish muscle are species-dependent [70] and influenced by the bioavailability in the environment. Low values have been attributed to the fact that muscle accumulates lower metals concentrations than the liver [71]. The trace elements in the marine environment (either introduced by natural runoff or from local mineral sources) are taken up by macroalgae, which in turn are consumed by various species. The fact that we found significant differences only in the fish species at the lowest trophic level (*K. vaigiensis*) suggests that those differences might be associated with differences in the geochemical dynamics between localities [50], probably determining short-term changes in cadmium contents in the fish diet. For example, it has been reported that cadmium concentrations in the brown algae *Padina durvillaei* increase after upwelling events [11].

Given the old mining activity associated with copper extraction in STR from the 19th to the mid−20th century, we expected copper levels to be the highest in fish caught in STR. This hypothesis was met for *K. vaigiensis*, but not for *B. polylepis* or *S. rectifraenum. *Copper content in the three fish species was several orders of magnitude lower than the mean Cu levels in the algae *P*. *durvillaei *(74 ± 29 µg g^−1^ and 82 ± 31 µg g^−1^) collected in the central and southern sites of STR, respectively [11], and similar to the levels reported for the scorpionfish *Scorpaena mystes* (0.7 ± 0.4 µg g^−1^) sampled in the same area [54]. These findings indicate that copper accumulated in algae is not fully assimilated by fish. Sediments from STR contain high copper and zinc levels. However, the high zinc levels recorded in coastal sediments (11,670 ± 10,800 µg g^−1^) [12] are not reflected in the levels observed in the algae *P*. *durvillaei *(63 ± 43 µg g^−1^) collected from these same sites; this finding has been attributed to the low geochemical mobility and bioavailability of zinc [11]. Zinc was the only element found at concentrations that were statistically different between the three species, being highest in *B. polylepis*, especially in specimens from STR. In a study carried out in Bahía Santa Maria on the Pacific coast of Baja California Sur that included the analysis of trace elements in homogenized samples of head, gut, and abdomen of 40 fish species, *B. polylepis* showed the highest zinc content (2.65 µg g^−1^) [36]. Zinc levels as high as 256.8 ± 10.88 µg g^−1^ have been reported for this species [35]. In contrast, carnivorous species from the Gulf of California such as *Lutjanus argentiventris* and *Haemulopsis leuciscus* have recorded lower zinc (4.71 ± 0.22 µg g^−1^ and 3.81 ± 0.08 µg g^−1^, respectively) and copper (2.72 ± 0.29 µg g^−1^ and 2.80 ± 0.18 µg g^−1^, respectively) contents relative to herbivorous species [72]. Copper levels are lower in *K. vaigiensis*, *S. rectifraenum*, and *B. polylepis* from LAP than in other organisms inhabiting the same area, and these three species yielded copper levels within the range reported for these same species in other studies (Table 1). All the above information suggests that these and other elements found in fish come from the intake of sediment particles and dissolved ions ingested incidentally when feeding ([36].

Anthropogenic discharges of nitrogen compounds have been associated with significant increases of δ^15^N in coastal ecosystems [1]. In the present study, δ^15^N was significantly related to zinc content in the three fish species; thus, nitrogen levels in these species are likely influenced by the presence of nitrogen-enriched discharges that may include zinc particulate matter [73]. This may be occurring mainly in STR, a locality where zinc levels in sediments can be as high as five-fold the concentration in non-polluted areas [12]. During the rainy season, these polluted sediments can be transported into the marine environment by runoff [73]. On the other hand, δ^15^N in fish muscle is strongly related to the species [50], similar to zinc; therefore, in the case of *K. vaigiensis* and *S. rectifraenum*, the relationship between δ^15^N and zinc is probably more closely associated with the species than with the environment [74]. Only *B. polylepis* showed significant differences in zinc levels related to the season of the year and locality. In contrast, the relationship between iron and δ^13^C suggests that iron may have a benthic origin [50], influenced by continental inputs on the site [75], rather than being associated with the species.

The feeding habits of a species can influence the accumulation of trace elements in its tissues [50]. Therefore, it was expected that fish classified either as omnivores feeding preferentially on animals or carnivores, like *S. rectifraenum* and particularly *B. polylepis*, would have higher metal levels than species described as herbivores or omnivores with a preference for plants, like *K. vaigiensis*. However, this expectation was not met in the present study, nor has it been confirmed in other similar studies [72,76]. *K. vaigiensis*, an herbivore, had similar cadmium, lead, and copper concentrations than *B. polylepis*, a species classified either as a carnivore or an omnivore [25,26]. Zinc was the only metal with a higher concentration in *B. polylepis* than in *K. vaigiensis*, which is contrary to the findings reported in other trophic webs where carnivorous fish have shown lower zinc contents [72]. The omnivorous species *S. rectifraenum* [24] showed significantly higher cadmium and lead levels than *K. vaigiensis*, whereas copper and zinc concentrations were significantly higher in *K. vaigiensis*. The amount of a given metal accumulated in an aquatic organism depends on the particular element and the species involved and may vary across species according to the particular ecological niche, swimming behavior, and metabolic activity of each [77]. Most of the more evolved invertebrates such as mollusks and echinoderms, as well as all vertebrates, have proteins called metallotioneins, the amount of which depends on the species and physical activity of organisms [78]. These proteins, of low molecular weight and with a high content of amino and sulfhydryl groups, play a central role in the homeostasis of essential and divalent elements, mainly zinc.

However, other non-essential elements such as cadmium compete for the functional groups in these proteins. By displacing zinc from these proteins, cadmium becomes non-bioavailable; thus, metallothioneins also act as detoxifiers by decreasing the bioavailability of non-essential elements, thereby reducing their toxicity [79]. Accordingly, only the fraction of a trace element that is bioavailable in an organism will be biomagnified in its predators [66]. Therefore, mollusks and echinoderms may accumulate a higher amount of cadmium relative to the fish feeding on them [80]. This has been documented in previous studies where *Balistes polylepis* specimens showed a higher cadmium content (2.57 µg g^−1^ and 2.94 µg g^−1^) than their predators: striped marlin *Kajikia audax* (0.63 ± 0.70 µg g^−1^) and *Makaira nigricans* (0.29 ± 0.18 µg g^−1^), all of them captured at the southern tip of the Gulf of California [37].

Iron is an essential element, and thus no maximum permissible levels have been set in regulations. This element was found within the range of concentrations reported for these same species in other studies, which is consistent with the direct relationship between the carbon isotope ^13^C and iron content in organisms revealed by the MRL. Therefore, iron content is likely associated with benthic feeding habits, so that food may also involve the incidental intake of organic and inorganic matter from the substrate and water, which may be influenced by the composition of sediments and rocks [36]. The contents of iron in *K. vaigiensis* and copper in *S. rectifraenum* decrease in the rainy season, especially in STR. This pattern is likely associated with the content of these elements in their diet (macroalgae in both species), as the species composition and abundance of macroalgae change throughout the year [14], as well as the feeding habits of fish [54].

In addition to the biogeochemical characteristics of a site and the feeding habits of a species, the level of metals bioaccumulated in an organism may also depend on its body size. Canli and Atli [77] observed an inverse relationship between the size of *Atherina hepsetus* specimens and copper, lead, and iron contents. This inverse relationship between body size and metal content was observed in *S. rectifraenum* in the present study, as larger specimens of this species lower copper level. However, for *K. vaigiensis* and *S. rectifraenum,* a direct relationship was found between both length and lead and iron content, contrasting with the inverse relationships reported by [72] for other fish species inhabiting the Gulf of California.

Canli and Atli [77] reported that *Sparus auratus* showed no relationship between metal content and body size, which coincides with our findings for *B. polylepis* in the present study. Canli and Atli [77] also observed an inverse relationship between cadmium content and body size in *Mugil cephalus*, which was not recorded in any of the species included in the present study. Although copper content was inversely related to body size in *S. rectifraenum*, no significant differences were observed between sites for this species. This result indicates that copper content in *S. rectifraenum* may be more closely related to its metabolism and indicates the ability of this species to regulate this element even in copper-rich marine environments as STR. Other studies conducted in polluted sites where copper levels were expected to be high in sciaenidae fish [67,81], showed that copper remained within normal levels in this species, as found in this study. It is concluded that the variations of this element in some fish species can be attributed more to their physiology than to the influence of the local environment in the area where they live [67,81]. The residual concentration of an element in whole-soft tissues results from the mass balance among uptake of the element through water or food, its elimination through feces or pseudo-feces, and retention by physiological/chemical process, some of them considered as defense detoxifying mechanisms in which are involved thiol compounds such as metallothionein and glutathione [45,79,82].

### Health Risk Assessment

The muscle is an edible part of marine organisms that is commonly consumed by man; its consumption can be nutritionally important as fish contain unique long-chain polyunsaturated fatty acids (LC-PUFAs) and highly bioavailable essential micronutrients—vitamins (B and D) and minerals such as calcium, phosphorus, iodine, zinc, iron, and selenium [83]. However, it can also be hazardous if metal content in muscle is higher to the concentration required for human physiology. *K. vaigiensis* and *B. polylepis* are species of commercial importance for human consumption, while *S. rectifraenum* is an aquarium fish [15]. Cadmium concentrations in the fish species studied here were above the food safety thresholds established by the European Commission regulation for Cd in fish, i.e., 0.05 μg g^−1^ wet weight (0.25 μg g^−1^ dry weight [69]). Lead levels in *S rectifraenum* from STR were above the *Codex alimentarius* threshold of 0.3 μg g^−1^ wet weight (1.5 μg g^−1^ dry weight [69]) for Pb in fish [84]. The ingestion rate is the main factor that influences health risk associated with a specific food [85]. In the present study, a daily consumption of 30 g (210 g weekly) of *K. vaigiensis* or *B. polylepis* does not represent a health risk for human, in contrast, *S. rectifraenum* could be dangerous if consumed. The fishing site is a key factor. For example, the concentration of lead in *S. rectifraenum* collected in SRL and reported for other sites within the Gulf of California [54,71,76] is significantly lower than what we found in La Paz. Lead is an element that is causing a raising concern, as a dietary intake in adults of 1.50 μg/kg b.w. per day and 0.63 μg/kg b.w. per day produce cardiovascular and kidney affections, respectively [3]. 

Except for few cases, such as *Mugil cephalus* from the eastern coast of the Gulf of California [76], the cadmium levels in the edible portion of fish from the Gulf of California do not represent a risk for human health. On the other hand, the consumption of *K. vaigiensis* and *B. polylepis* contributes nutritionally an important amount of copper, zinc, and iron that do not represent a risk for human health. 

## 5. Conclusions

The mining activities examined in this study are not directly reflected on the metal contents in fish, instead of geochemical dynamics and fish physiology. Fish species in the same trophic level and inhabiting the same sites showed significant differences in the levels of the same elements; on the other hand, no differences in metal content were observed between species from different trophic levels. Each element accumulates differently in individual species.

According to this report, there is currently no risk of heavy metals toxicity derived from the human consumption of fish of the species and sites studied, except for lead in LAP. Further research is needed to assess the bioavailability of trace elements in different marine species to understand the mechanisms underlying bioaccumulation and assess the ecotoxicological risk of a site. Meanwhile, it is important to maintain permanent seafood safety monitoring programs in place. Also, as a rule of thumb, the consumption of a diverse diet (different fish species, from different locations) may reduce the risk of food safety problems.

## Figures and Tables

**Figure 1 ijerph-18-00844-f001:**
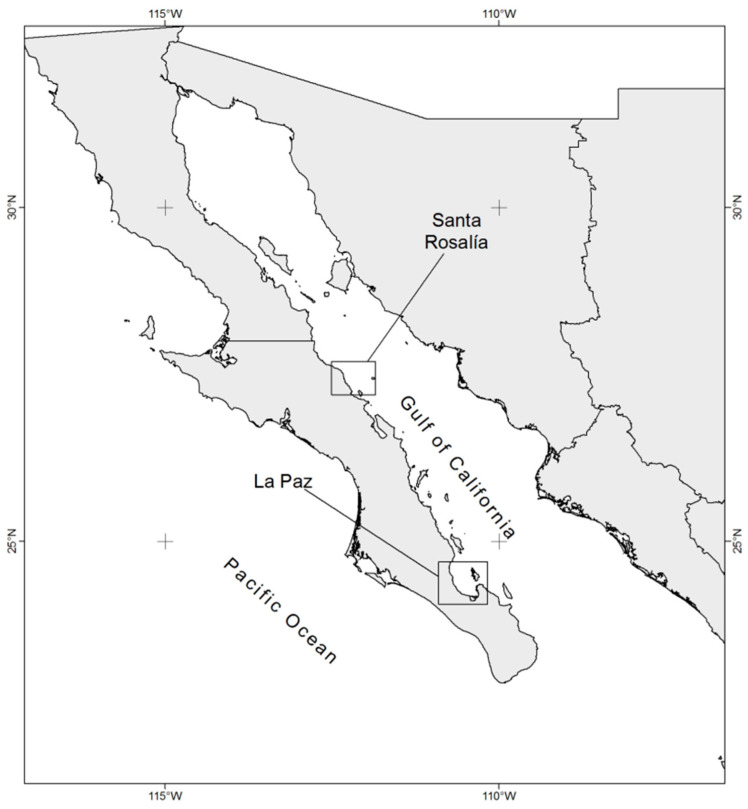
Sampling sites of *Kyphosus vaigiensis*, *Stegastes rectifraenum*, and *Balistes polylepis* in the vicinity of Santa Rosalía (STR) and Bahía de La Paz (LAP).

**Figure 2 ijerph-18-00844-f002:**
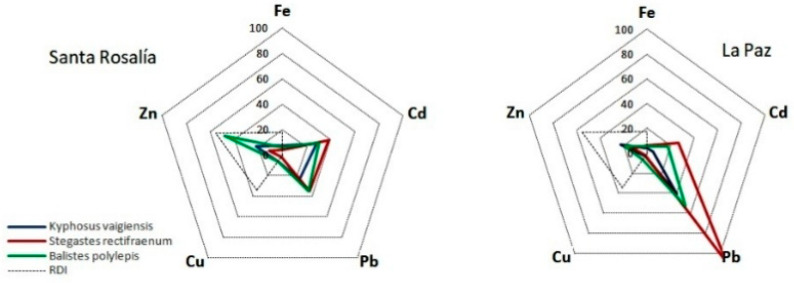
Estimated daily intake (EDI) of copper (Cu), zinc (Zn), iron (Fe), cadmium (Cd), and lead (Pb) in humans from the consumption of muscle of *K. vaigiensis*, *S. rectifraenum* and *B. polylepis* fish species in relation to the recommended daily dose (RDI) and how it last represents versus the reference dosis of each element.

**Table 1 ijerph-18-00844-t001:** Concentration (average ± standard deviation or minimum and maximum) of cadmium, lead, copper, zinc, and iron (μg g^−1^ dry weight) in muscle of fish of the genera *Kyphosus*, *Stegastes*, and *Balistes*.

Species	*n*	Size (cm)	Cadmium	Lead	Copper	Zinc	Iron	Locality	Reference
***Kyphosus***									
***K. cinerascens***	-	35–42	-	-	0.63–1.1	16.0–41.8	-	Great Barrier Reef, Australia	[29]
***K. cinerascens ****	14	-	3.56 ± 0.10 **	23.21 ± 0.72 **	1.13 ± 0.017	6.88 ± 0.08	22.99 ± 0.28	South China Sea	[30]
***K. vaigiensis (as K. lembus) ****	15	-	1.57 ± 0.04 **	16.43 ± 0.46 **	0.68 ± 0.012	5.23 ± 0.12	16.02 ± 0.22	South China Sea	[30]
***K. vaigiensis (as K. lembus)***	-	-	0.016	0.039	1.549	33.860	23.060	Spratly islands, China	[31]
***K. vaigiensis***	5	31.7 ± 5.0	-	<0.08	<0.83	24.1 ± 8.51	8.69 ± 1.06	New Caledonia	[32]
***K.* sp.**	-	-	0.50	4.00	0.80	28.30	6.00	Gulf of Aqaba, Red Sea	[33] ^1^
***K.* sp.**	3	29	0.002	0.017	-	0.97	-	Caribbean Sea, Colombia	[34]
***K. vaigiensis***	42	31.82 ± 1.32	0.18 ± 0.03	0.59 ± 0.16	1.60 ± 0.20	42.45 ± 3.01	33.24 ± 3.67	Gulf of California, Santa Rosalía	Present study
***K vaigiensis***	27	23.70 ± 1.42	0.03 ± 0.01	1.02 ± 0.39	0.83 ± 0.19	43.39 ± 3.30	18.24 ± 2.58	Gulf of California, Bahía de La Paz	Present study
***Stegastes***									
***S. fasciolatus***	3	NA	0.8–1.7	13.00–16.00	-	-	-	Tern Island, North Pacific	[27]
***S. rectifraenum***	40	11.09 ± 0.16	0.24 ± 0.02	0.90 ± 0.22	0.53 ± 0.10	21.19 ± 0.42	18.51 ± 2.48	Gulf of California, Santa Rosalía	Present study
***S. rectifraenum***	28	12.47 ± 0.13	0.17 ± 0.06	2.76 ± 0.49	0.60 ± 0.10	23.88 ± 0.77	26.24 ± 4.90	Gulf of California, Bahía de La Paz	Present study
***Balistes***									
***B. capriscus***	-	-	2.615	0.995	0.013	-	0.253	Libya, Mediterranean Sea	[28]
***B. polylepis ****	524	-	5.282 ± 1.020	0.023 ± 0.001	1.29 ± 0.10	64.2 ± 2.72	-	Mazatlán, Mexico	[35] ^2^
***B. polylepis***	3	15.5	0.02	0.11	0.073	2.65	3.80	Pacific Baja California	[36]
***B. polylepis***	2	-	2.57 ± 1.03	-	-	-	-	Gulf of California	[37] ^3^
***B. polylepis***	2	-	2.94 ± 0.41	-	-	-	-	Gulf of California	[37] ^4^
***B. polylepis***	40	22.97 ± 0.89	0.19 ± 0.03	0.89 ± 0.25	1.55 ± 0.68	93.72 ± 14.77	27.97 ± 2.80	Gulf of California, Santa Rosalía	Present study
***B. polylepis***	28	26.62 ± 1.05	0.11 ± 0.05	1.33 ± 0.36	1.66 ± 0.71	36.16 ± 1.60	23.98 ± 7.80	Gulf of California, Bahía de La Paz	Present study

^1^ = phosphate-polluted area with urban anthropogenic impact; ^2^ = as part of the diet of *Istiophorus platypterus*; ^3^ = as part of the diet of *K. audax*; ^4^ = as part of the diet of *M. nigricans*. * = µg g^−1^ wet weight; ** = ng g^−1^; - = Not available.

**Table 2 ijerph-18-00844-t002:** Concentrations of metals (µg g^−1^; DW) in muscle of *K. vaigiensis*, *S. rectifraenum* and *B. polylepis* by location and by season. Data are shown as mean ± SE.

Location	Santa Rosalía			La Paz		
Season	Dry	Rainy	Annual Average	Dry	Rainy	Annual Average
***Kyphosus vaigiensis***						
cadmium	0.17 ± 0.05	0.20 ± 0.04	0.19 ± 0.03 ^A^	0.06 ± 0.03 ^a^	<0.01 ^b^	0.03 ± 0.01 ^B^
lead	0.58 ± 0.17	0.60 ± 0.27	0.59 ± 0.16	2.08 ± 0.71 ^a^	<0.07 ^b^	1.02 ± 0.39
copper	1.23 ± 0.26	1.92 ± 0.30	1.57 ± 0.20 ^A^	1.71 ± 0.21 ^a^	<0.017 ^b^	0.83 ± 0.20 ^B^
zinc	47.90 ± 5.27	36.01 ± 1.86	42.09 ± 2.96	55.01 ± 4.72 ^a^	32.60 ± 2.09 ^b^	43.39 ± 3.30
iron	47.70 ± 5.14 ^a^	16.78 ± 1.93 ^b^	32.60 ± 3.65 ^A^	13.30 ± 2.68	22.83 ± 4.04	18.24 ± 2.59 ^B^
***Stegastes rectifraenum***						
cadmium	0.20 ± 0.03	0.27 ± 0.03	0.24 ± 0.02	0.12 ± 0.04	0.21 ± 0.11	0.17 ± 0.06
lead	1.01 ± 0.31	0.80 ± 0.31	0.90 ± 0.22 ^A^	4.16 ± 0.43 ^a^	1.37 ± 0.71 ^b^	2.76 ± 0.49 ^B^
copper	1.05 ± 0.12 ^a^	0.05 ± 0.03 ^b^	0.53 ± 0.10	0.88 ± 0.14 ^a^	0.33 ± 0.10 ^b^	0.60 ± 0.10
zinc	20.97 ± 0.63	21.39 ± 0.57	21.19 ± 0.42 ^A^	24.11 ± 0.94	23.66 ± 1.26	23.88 ± 0.77 ^B^
iron	17.16 ± 2.62	19.72 ± 4.14	18.51 ± 2.48	17.53 ± 2.63	34.96 ± 9.02	26.24 ± 4.90
***Balistes polylepis***						
cadmium	0.14 ± 0.05	0.21 ± 0.04	0.19 ± 0.03	0.09 ± 0.04	0.14 ± 0.09	0.11 ± 0.05
lead	1.91 ± 0.56 ^a^	0.31 ± 0.14 ^b^	0.89 ± 0.25	1.44 ± 0.48	1.23 ± 0.56	1.33 ± 0.36
copper	1.49 ± 0.89	1.59 ± 0.95	1.55 ± 0.68	2.73 ± 1.33	0.59 ± 0.40	1.66 ± 0.71
zinc	41.55 ± 3.00 ^a^	123.54 ± 20.58 ^b^	93.72 ± 14.77 ^A^	33.71 ± 2.15	38.62 ± 2.25	36.16 ± 1.60 ^B^
iron	39.16 ± 3.86 ^a^	21.57 ± 3.07 ^b^	27.97 ± 2.80	11.96 ± 2.86	29.13 ± 10.82	23.98 ± 7.80

Different letters indicate significant differences (*p* < 0.05) in annual average between locations (Santa Rosalia ^(A)^ vs. La Paz ^(B)^) and between seasons within each location—dry ^(a)^ vs. Rainy ^(b)^.

**Table 3 ijerph-18-00844-t003:** General characteristics, body length (cm), metal concentrations (µg g^−1^; dw), *δ*^15^N and *δ*^13^C (‰) in muscle of three fish species. Data are shown as mean ± SE (minimum–maximum).

Scientific Name	*Kyphosus vaigiensis*	*Stegastes rectifraenum*	*Balistes polylepis*
Fish Photo	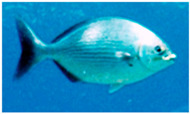	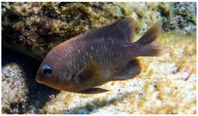	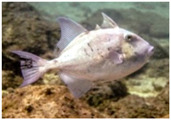
Common name	Blue-bronze chub/Brassy chub	Cortez damselfih	Finescale triggerfish
Common local name	Chopa rayada	Jaqueta de Cortés	Cochi/Cochito
Commercial value [58]	1.5 USD/k—artisanal fisheries [59,60]	18 USD/k—ornamental reef fishery [61,62]	1.3 USD/k—artisanal fisheries [59]
Feeding habits	Herbivore [17]	Omnivore [35,63]	Carnivore [25]
Trophic level [64]	2.0 ± 0.00	2.0 ± 0.00	3.3 ± 0.37
Sample size	70	68	61
Lenght (cm)	28.69 ± 1.08 ^a^ (14.30–54.00)	11.66 ± 0.14 ^b^ (8.50–14.20)	24.64 ± 0.71 ^a,b^ (15.30–38.50)
cadmium	0.13 ± 0.02 ^a^ (<0.01–0.68)	0.21 ± 0.03 ^b^ (<0.01–1.0)	0.15 ± 0.03 ^a,b^ (<0.01–1.00)
lead	0.75 ± 0.18 ^a^ (<0.04–7.41)	1.67 ± 0.26 ^b^ (<0.04–6.45)	1.09 ± 0.22 ^a,b^ (0.04–6.04)
copper	1.28 ± 0.15 ^a^ (<0.017–5.19)	0.56 ± 0.07 ^b^ (<0.017–1.95)	1.60 ± 0.49 ^a,b^ (<0.017–20.50)
zinc	42.59 ± 2.21 ^a^ (22.34–114.22)	22.30 ± 0.43 ^b^ (17.17–33.08)	67.30 ± 8.79 ^c^ (23.34–217.64)
iron	27.06 ± 2.58 ^a^ (2.60–103.34)	21.69 ± 2.51 ^a^ (2.09–141.15)	26.46 ± 3.16 ^a^ (0.64–146.76)
δ^13^C	−13.66 ± 0.26 ^a^ (−20.13–−9.75)	−14.99 ± 0.19 ^b^ (−18.39–−12.31)	−16.25 ± 0.21 ^c^ (−18.46–−11.92)
δ^15^N	16.77 ± 0.16 ^a^ (13.33–18.57)	18.12 ± 0.13 ^b^ (15.99–19.91)	17.73 ± 0.17 ^b^ (14.03–19.52)

Different letters (^a, b^ or ^c^) after the means ± error standard indicate significant differences between fish species (*p* < 0.05) meanwhile, equal letters means no significant differences (*p* > 0.05) among species. The letters are not related to a specific species. The content of a trace element in the three species can be equal or different among them (sharing or not the same letter).

**Table 4 ijerph-18-00844-t004:** Multiple categorical regression results.

Statistics	Cadmium			Lead			Cooper			Zinc			Iron		
	Coeff	SE	t-Value	Coeff	SE	t-Value	Coeff	SE	t-Value	Coeff	SE	t-Value	Coeff	SE	t-Value
Intercept	−6.75	2.65	−2.55 *	−1.35	2.86	−0.47	−0.33	3.48	−0.10	1.24	0.64	1.95	5.31	1.21	4.37 *
Location__STR_	1.59	0.30	5.37 *	−0.19	0.32	−0.60	1.43	0.39	3.69 *	0.08	0.07	1.17	0.40	0.14	2.86 *
Season__Rainy_	0.07	0.30	0.23	−1.30	0.33	−3.98 *	−2.12	0.40	−5.33 *	0.26	0.09	3.55 *	−0.18	0.14	−1.34
Sp_*_K. vaigiensis_*	0.05	0.37	0.14	−0.87	0.40	−2.17 *	0.59	0.49	1.20	−0.10	0.11	−1.13	−0.25	0.17	−1.48
Sp_*_S. rectifraenum_*	0.14	0.48	0.29	2.09	0.52	4.05 *	−0.25	0.63	−0.39	−0.56	0.01	−4.90 *	0.29	0.22	1.28
L_std	−0.03	0.02	−1.49	0.11	0.02	4.67 *	0.01	0.03	0.34	0.02	0.01	2.99 *	0.03	0.01	2.50 *
δ^13^C	−0.05	0.09	−0.61	−0.11	0.09	−1.16	−0.17	0.11	−1.51	−0.07	0.02	−3.24	0.01	0.04	0.22
δ^15^N	0.15	0.13	1.15	−0.23	0.14	−1.68	−0.22	0.17	−1.33	0.06	003	1.91	−0.17	0.06	−2.91 *
Model R^2^	0.17			0.22			0.22			0.44			0.07		
Model *p*<	8.07e^−10^			1.25e^−10^			1.73e^−11^			<2.2e^−16^			9.56e^−4^		

For the categorical variables (location, Season, Specie—Sp) the subindex indicates the category with higher explanatory weight when the coefficient (Coeff) takes positive value. The relative weight of the other category within that coefficient is higher for negative values. In the case of Sp, because dummy variables were used, the first (Sp_*_K. vaigiensis_*) is positive for *Kyphosus vaigiensis* and negative for the others combined, and second (Sp_*_S. rectifraenum_*) is positive for *Stegastes rectifraenum* and negative for *Balistes polylepis.* SE: standard error. STR: Santa Rosalia; L_std: length standard. Asterix (*) indicate significant contribution at *p* < 0.05. Reported R^2^ and *p* values correspond to the fitted model (using significant contributing variables only).

## Data Availability

The data presented in this study are available on request from the corresponding author. The data are not publicly available due the data is been used by students.

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
