# Peer review of "Cadmium, Lead, Copper, Zinc, and Iron Concentration Patterns in Three Marine Fish Species from Two Different Mining Sites inside the Gulf of California, Mexico"

_ijerph, 2021, doi:10.3390/ijerph18020844_

Round 1

Reviewer 1 Report

Below are specific comments on the manuscript:

Abstract line 24. Lack of concentrations registered in the fish species.

Introduction, line 112: It should be mentioned here if the fish species are sedentary and accumulation of metals are likely to result from local availability of those elements 

Materials and Methods, line 97. The reference Trumbo et al, 2001 is missing

Results, section 3.2. Information on the biometric parameters and the number of specimens with values below detection limits should be first in Results. 

Results, section 3.2, line 17. It is surprising that such a number of specimens showing Cu concentration below 0.017 mg/kg. It should be compared check th calculations and compared with the literature

Discussion, line 90. Some of the questions presented by the authors are well explained with the following approach: residues of an element in whole-soft tissues (a single box approach) is the mass balance among uptake of the element (through water or food), elimination (feces or pseudo-feces) and retention by physiological/ chemical processes, some of them considered as defense detoxifying mechanisms, such as the low-molecular proteins and Cd retention.

Author Response

Thank you for allowing improves our manuscript and made it more understandable and informative.

 Point 1. Abstract line 24. Lack of concentrations registered in the fish species.

Response 1: Attended. We modified the abstract to:

“The highest concentrations recorded were cadmium (0.21 ± 0.03 µg g−1; dw) and lead (1.67 ± 0.26 µg g−1; dw), in S. rectifraenum; copper (1.60 ± 0.49 µg g−1; dw) and zinc (67.30 ± 8.79 µg g−1; dw), in B. polylepis; and iron (27.06 ± 2.58 µg g−1; dw), in K. azureus.”

Point 2. Introduction, line 112: It should be mentioned here if the fish species are sedentary and accumulation of metals are likely to result from local availability of those elements 

Response 2:  Attended. We included a sentence and a citation at lines 121-123 within the introduction:

“The fact that most records of these three fish species are from rocky inshore reefs dominated by Sargassum along the Gulf of California [15], suggests that their accumulation of metals is linked to local conditions.”

Point 3. Materials and Methods, line 97. The reference Trumbo et al, 2001 is missing

Response 3: Attended, citation included, and references double checked.

Point 4. Results, section 3.2. Information on the biometric parameters and the number of specimens with values below detection limits should be first in Results. 

Response 4: Attended, section complemented and moved.

Point 5. Results, section 3.2, line 17. It is surprising that such a number of specimens showing Cu concentration below 0.017 mg/kg. It should be compared check the calculations and compared with the literature.

Response 5: Attended. Data were checked, and a paragraph was included in the discussion

“Undetectable or very low concentrations of copper (<0.03 µg g-1) in muscle have been previously reported (67-69). Concentrations of this and other elements in fish muscle are species-dependent (70) and influenced by the bioavailability in the environment. Low values have been attributed to the fact that muscle accumulates less metals than the liver (71).”

Point 6. Discussion, line 90. Some of the questions presented by the authors are well explained with the following approach: residues of an element in whole-soft tissues (a single box approach) is the mass balance among uptake of the element (through water or food), elimination (feces or pseudo-feces) and retention by physiological/ chemical processes, some of them considered as defense detoxifying mechanisms, such as the low-molecular proteins and Cd retention.

Response 6: Thanks, we agree, and the idea was included at the end of the first part of the discussion.

Reviewer 2 Report

The proposal is of scientific and public interest. I have only a few observations.

The map is confusing. Only the upper map is clear.

Authors could extend the public health context, since the species and study area are described very well, but little about health risk.

Table 3. The images are not the best to describe the species, especially that of Stegastes rectifraenum.

In the bibliography it is found that some species names are not written in italics, please check.

i.e lines 198, 240, 272, 298.

Author Response

Thank you for allowing improves our manuscript and made it more understandable and informative. Please, find below our responses.

Point 1. The map is confusing. Only the upper map is clear

Response 1:  Attended. The map was simplified for clarity.

 Point 2. Authors could extend the public health context, since the species and study area are described very well, but little about health risk.

Response 2: Reply: Attended: We included an improved section on health risk at the last part of the discussion.

Point 3. Table 3. The images are not the best to describe the species, especially that of Stegastes rectifraenum.

Response 3: Attended, we substituted the image for S. rectifraenum and improved the images quality.

Point 4. In the bibliography it is found that some species names are not written in italics, please check.

i.e. lines 198, 240, 272, 298.

Response 4: Attended. All references were revised, and species names were written in italics.

Reviewer 3 Report

This manuscript deals with the presence of some metals in three different fish species collected at two minig site in Gulf of California. Moreover a food safety assessment was performed. The topic is interesting and overall the experimental design was performed correctly. Methodology of labortory analysis was correct but I have concerns about statistical analyses. Hereafter my comments:

1) Figure 1 needs to be moved below in M&M section

2) In introduction you have to state the expectation of your work. Moreover, at the end of the introduction I suggest to rephrase the sentence indicating that your work is the first one in the area. As you stated in the discussion section previous works monitored metal contamination in fish of the same study area. Please also explain why it is important to consider in data analysis the time of year, fish size and trophic niche.

3) In M&M some details are necessary:

  • how many fish per species did you collect in each sampling location?
  • did you consider both the sexes or did you planned analyses of males and females?
  • how many analytical replicates did you perform per sample?
  • You indicated the LOD. Did you calculated the LOQ for each metal? Are LOD and LOQ similar for the three fish species?

4) What are the variables included in the statistical models you performed? You stated that ANOVA was performed. Did you mean a factorial ANOVA including time of year, sampling location and species in the models? Did you consider the interaction effect among factors?

You used two different statistical approach. You have to justify your choice and to explain the additional information given by the application of multiple categorical regrassion rather than ANOVA.

5) in the result section all the statistical indicators ara missing. This is particularly true for ANOVA, although I did not understand when you applied ANOVA. F value and degree of freedom need to be added. You could add a Table reporting the results of ANOVA statistical analysis as you reported for multiple categorical regression.

6) how can you explain that for a notable amount of fish you did not measure metals? Moreover, in which locations did you not detect metals? Please also report the frequency of detection of metals in fish species. How did you calculated the annual average when you did not detect metals in a specific time of year?

7) figure 2 is not clear and needs to be implemented. What is the added value of this figure compared to the description in the text?

Author Response

Thank you for allowing improves our manuscript and made it more understandable and informative. Please, find below our responses.

Point 1.  Figure 1 needs to be moved below in M&M section

Respond 1:  Attended. Moved to 2.1

Point 2. In introduction you have to state the expectation of your work. Moreover, at the end of the introduction I suggest to rephrase the sentence indicating that your work is the first one in the area. As you stated in the discussion section previous works monitored metal contamination in fish of the same study area. Please also explain why it is important to consider in data analysis the time of year, fish size and trophic niche.

Respond 2: Attended, we included the phrase “…two different mining facilities, one for copper and another for phosphorite, and to test whether the type of mining determines the metal concentration in fish muscle.” To state our original expectations. Also, we rephrased the following sentence to “To our knowledge, this is the first report comparing different mining activities in the area.” , following the reviewers suggestion. Finally, we stated the importance of other factors through the sentence “To evaluate the relative importance of the type of mining activity in determining the metal concentrations in fish muscle, we also evaluated other factors, including the time of the year, the trophic niche, and the fish size.”

Point 3. In M&M some details are necessary:

  • how many fish per species did you collect in each sampling location?

Respond: Attended. The number of fish per species in each sampling location was specified at the second paragraph of section 2.2 as:

“A total of 70 specimens of Kyphosus vaigiensis (43 from STR; 27 from LAP), 68 of Stegastes rectifraenum (40 from STR; 28 from LAP), and 61 of Balistes polylepis (33 from STR; 28 from LAP) were captured and transported on ice to the laboratory.”

  • did you consider both the sexes or did you planned analyses of males and females?

Respond:  We did not consider fish sex as a factor within our experimental design. Now that we learned that fish physiology plays a major role in determining the metal concentration in muscle, we will include it within our future research.

  • how many analytical replicates did you perform per sample?

Respond: Attended. Each sample was tested in triplicate. This is now specified in M&M, section 2.3.

  • You indicated the LOD. Did you calculated the LOQ for each metal? Are LOD and LOQ similar for the three fish species?

Respond: Attended. LOD and LOQ were specified in the methods, section 2.3.

“The detection limits/quantification limits (µg g−1) were as follows: Cd: 0.01/0.02; Pb: 0.07/0.10; Cu: 0.017/0.020; Zn: 0.021/0.060; Fe: 0.65/1.35.”

Point 4. What are the variables included in the statistical models you performed? You stated that ANOVA was performed. Did you mean a factorial ANOVA including time of year, sampling location and species in the models? Did you consider the interaction effect among factors?

Respond: Attended. We improved the description of the ANOVA analyses.

One-way ANOVA analyses were used to assess significant differences in metal concentrations between location (Santa Rosalía, La Paz) and between seasons per location (dry, rainy) for each fish species and significant differences in length, metal concentration and stable isotopes between species (K. vaigiensis, S. rectifraenum, B. polylepis) followed by the post hoc Tukey test (51).”

-You used two different statistical approach. You have to justify your choice and to explain the additional information given by the application of multiple categorical regression rather than ANOVA.

-Respond: Attended. We included the sentence:

“The multiple categorial regression was applied to evaluate and compare the relative weight of each of the factors in determining the concentration of the different elements in fish muscle. “within section 2.6.

Point 5.  in the result section all the statistical indicators are missing. This is particularly true for ANOVA, although I did not understand when you applied ANOVA. F value and degree of freedom need to be added. You could add a Table reporting the results of ANOVA statistical analysis as you reported for multiple categorical regression.

Respond 5: Statistical indicators were specified for the ANOVA.

Point 6. How can you explain that for a notable amount of fish you did not measure metals? Moreover, in which locations did you not detect metals? Please also report the frequency of detection of metals in fish species. How did you calculated the annual average when you did not detect metals in a specific time of year?

Respond 6: Attended. The two studied areas have historical mine activities, and virtually no industrial or agricultural activities. The bioavailability of the elements is an important factor to be bioaccumulated that can be affected by human activities or other environmental conditions. It is important to note that the analysis was done in the muscle, which accumulates lower metals concentration than the liver (71), this is now included in the discussion. Another important fact is that most of the articles reporting metals use mean± std error or deviation, instead the concentrations range, and therefore the proportion of undetectable values is unclear and do not allow for intercomparisons.

We included a section at the beginning of the results:

“Of the 70 specimens of K. vaigiensis, 38 (17 from STR, 21 from LAP) showed undetectable cadmium levels (<0.01 µg g−1), 48 (27 from STR, 21 from LAP) had undetectable lead levels (<0.07 µg g−1), and 16 (1 from STR, 15 from LAP) showed undetectable copper levels (<0.017 µg g−1). Of the 68 samples of S. rectifraenum, 20 (3 from STR, 17 from LAP) showed undetectable cadmium levels, 34 (23 from STR, 11 from LAP) had undetectable lead levels, and 26 (19 from STR, 7 from LAP) showed undetectable copper levels. Of the 61 B. polylepis organisms, 24 (7 from STR, 17 from LAP) showed undetectable cadmium levels, 48 (27 from STR, 21 from LAP) had undetectable lead levels, and 18 (5 from STR, 13 from LAP) showed undetectable levels of copper.”

Also at 2.5:

“Before the statistical tests, to each sample that presented an undetectable concentration of some metal it was assigned a value that corresponds to half the detection limit of such metal [49].”

Point 7.  Figure 2 is not clear and needs to be implemented. What is the added value of this figure compared to the description in the text?

Respond 7: We believe the figure is useful and not redundant with the text. In health risk sections, Radial plots were built to compare the EDI for each of the metals, species, and locations to their corresponding RfD or the total daily intake. This graphics gives the opportunity to see in an integrative way all the comparations EDI vs RfD and Daily intake of all the metals analyzed by location. We modified the figure legend.

Round 2

Reviewer 3 Report

The authors improved satisfactorily the manuscript according to my suggestions. However I have a concern on the statistical analysis. You stated that: "One-way ANOVA
analyses were used to assess significant differences in metal concentrations between location (Santa Rosalía, La Paz) and between seasons per location (dry, rainy) for each fish species and significant differences in length, metal concentration and stable isotopes between species..". I think that a one-way ANOVA is not appropriate to assess statistical differences between season per location. One-way ANOVA means that the predictor is one (i.e., location or fish species or season) but if you would like to check for differences in metal levels for the interaction 'season x location' you have to perfrome a two-way or factorial ANOVA, including both the predictors in the model.

I also suggest to include the exact P value associated to F values for all the test you cited within the text.

Author Response

Comment Reviewer 3:

Point 1. The authors improved satisfactorily the manuscript according to my suggestions. However I have a concern on the statistical analysis. You stated that: "One-way ANOVA analyses were used to assess significant differences in metal concentrations between location (Santa Rosalía, La Paz) and between seasons per location (dry, rainy) for each fish species and significant differences in length, metal concentration and stable isotopes between species..". I think that a one-way ANOVA is not appropriate to assess statistical differences between season per location. One-way ANOVA means that the predictor is one (i.e., location or fish species or season) but if you would like to check for differences in metal levels for the interaction 'season x location' you have to perfrome a two-way or factorial ANOVA, including both the predictors in the model. I also suggest to include the exact P value associated to F values for all the test you cited within the text.

Response 1: Thank you for your comments and suggestions. The exact P-value associated to F-values for all the tests cited within the text were included (page 9: lines 166 to 190; page 10: lines: 201-202; page 12: lines: 5 to 7).  Regarding the ANOVA, a one-way ANOVA was considered because the objective was to compare seasons within each location (one predictor) for each fish species, not for the interaction  'season x location'. This was specified in methods (line 94-98):

"One-way ANOVA analyses were used to assess significant differences in metal concentrations between location (Santa Rosalía, La Paz), also between seasons (dry, rainy) within each location and ....." 

Additionally, the multiple categorial regressions were applied to evaluate and compare the relative weight of each of the factors in determining the concentration of the different elements in fish muscle.
